# Exploring Localized Provoked Vulvodynia: Insights from Animal Model Research

**DOI:** 10.3390/ijms25084261

**Published:** 2024-04-11

**Authors:** Yara Nakhleh-Francis, Yaseen Awad-Igbaria, Reem Sakas, Sarina Bang, Saher Abu-Ata, Eilam Palzur, Lior Lowenstein, Jacob Bornstein

**Affiliations:** 1Department of Obstetrics and Gynecology, Galilee Medical Center, Nahariya 2210001, Israel; sarinabang88@gmail.com (S.B.); liorl@gmc.gov.il (L.L.); mdjacob@gmail.com (J.B.); 2Research Institute of Galilee Medical Center, Nahariya 2210001, Israel; yaseenawad123@gmail.com (Y.A.-I.); reem_sakas@hotmail.com (R.S.); saher9644@gmail.com (S.A.-A.); eilam.palzur@gmail.com (E.P.); 3Azrieli Faculty of Medicine, Bar-Ilan University, Safed 1311502, Israel

**Keywords:** Localized Provoked Vulvodynia (LPV), chronic vulvar pain, vulvodynia, animal model of vulvodynia, vulvodynia induction, hyperinnervation

## Abstract

Provoked vulvodynia represents a challenging chronic pain condition, characterized by its multifactorial origins. The inherent complexities of human-based studies have necessitated the use of animal models to enrich our understanding of vulvodynia’s pathophysiology. This review aims to provide an exhaustive examination of the various animal models employed in this research domain. A comprehensive search was conducted on PubMed, utilizing keywords such as “vulvodynia”, “chronic vulvar pain”, “vulvodynia induction”, and “animal models of vulvodynia” to identify pertinent studies. The search yielded three primary animal models for vulvodynia: inflammation-induced, allergy-induced, and hormone-induced. Additionally, six agents capable of triggering the condition through diverse pathways were identified, including factors contributing to hyperinnervation, mast cell proliferation, involvement of other immune cells, inflammatory cytokines, and neurotransmitters. This review systematically outlines the various animal models developed to study the pathogenesis of provoked vulvodynia. Understanding these models is crucial for the exploration of preventative measures, the development of novel treatments, and the overall advancement of research within the field.

## 1. Introduction

Vulvodynia, characterized as a chronic vulvar pain syndrome, affects an estimated 7–28% of women, significantly undermining their quality of life. This condition is intricately linked with profound psychosocial implications, including depression, anxiety, and a noticeable decline in sexual desire. The most frequently encountered subtype, Localized Provoked Vulvodynia (LPV), is defined by pain and hypersensitivity localized to the vestibule, persisting for at least three months. Such discomfort is typically provoked by both sexual and non-sexual activities [1].

The etiology of vulvodynia is recognized as complex and multifactorial, involving an array of biomedical factors such as abnormalities in vestibular mucosa, issues with pelvic floor musculature, and dysregulation of pain, alongside psychosocial factors including distress and childhood traumas [2,3,4,5,6]. Despite the expanding body of research in this area, the etiopathogenesis of vulvodynia remains largely elusive [3,7,8,9].

Animal models have emerged as a valuable tool in bridging gaps in our understanding of vulvodynia, addressing limitations inherent in human studies, such as the restricted ability to examine tissues and the complexities involved in assessing pain perception. These models are instrumental in the investigation of potential new treatments and preventive measures. Moreover, they offer significant insights into the histopathological and clinical developments at different stages of the condition [10].

In recent years, significant progress has been made in the development of animal models for the study of vulvodynia. These models are essential for understanding the mechanisms underlying this condition and can be broadly classified into three main categories based on the method of induction:Inflammation-based models, where vulvodynia is induced using agents such as Candida, zymosan, and Complete Freund’s Adjuvant (CFA), highlighting the role of inflammatory processes in the development of vulvodynia [1,11,12,13,14,15,16,17,18].Allergy-induced models, which utilize allergic reactions to substances like oxazolone (OX) or methylisothiazolinone (MI) to mimic the condition, shedding light on the potential involvement of allergic responses in vulvodynia [16,17,18,19].Hormonal exposure models, where early exposure to hormones is used to induce vulvodynia, suggesting the influence of hormonal factors on the condition’s pathogenesis [9].

This narrative review aims to comprehensively present and discuss the animal models employed in vulvodynia research, comparing the various induction methods and analyzing their findings. The goal is to identify promising directions for future research, with a focus on the development of effective prevention strategies and treatment advancements.

## 2. Methods

### 2.1. Animals

To investigate vulvodynia, studies have incorporated both rats and mice, employing a variety of strains to achieve a nuanced understanding of the condition across different animal models. In line with established research norms, these animals were kept under a controlled 12:12-h light/dark cycle with unrestricted access to food and water, adhering to widely accepted standards [20]. Furthermore, all experimental procedures, encompassing habituation, injections, and behavioral assessments, were conducted following the guidelines endorsed by the National Institutes of Health.

Reflecting the genetic diversity seen in the human female population, the use of outbred species has been instrumental in these studies [21]. Conversely, inbred mice have been the focus of two studies [13,15], underscoring the preference for inbred species in genetic research to minimize variability in the results [22] (Table 1).

### 2.2. Methods for Inducing LPV Animal Models

A diverse array of methods and techniques has been applied to induce vulvodynia in animal models, as outlined in Table 2. These approaches range from injections of various substances to topical applications, each method carefully designed to effectively simulate the condition. The specific details of these induction methods, as utilized in the respective studies, are concisely summarized in Table 3.

### 2.3. Vulvar Allodynia and Hypersensitivity Measurements in LPV Animal Models

To assess vulvar allodynia and hypersensitivity in animal models, researchers have primarily utilized two methodologies: mechanical sensitivity testing and thermal sensitivity assessment.

### 2.4. Mechanical Sensitivity Testing in LPV Animal Models

This approach involves determining the vulvar mechanical pain threshold of the subjects. Different studies have adopted both manual and electronic techniques for this purpose, allowing for the precise quantification of the force required to elicit a pain response. These methods facilitate the evaluation of the degree of mechanical allodynia present in the animals (Table 4).

### 2.5. Thermal Sensitivity Assessment in LPV Animal Models

The hot-plate test, traditionally used to evaluate hind-paw hypersensitivity, has been adapted to study vulvodynia. In this context, rats displaying signs of vulvodynia have been observed to engage in behaviors indicative of discomfort, such as licking the vulvar area and executing escape maneuvers like jumping [11]. This adaptation of the hot-plate test to assess thermal sensitivity in the context of vulvodynia provides insights into the animal’s response to thermal stimuli, further elucidating the complexity of pain perception associated with this condition.

### 2.6. Assessment of Post-Induction Tissue Alteration in LPV Animal Models

#### 2.6.1. Histopathological Examination in LPV Animal Models

After sacrification, the examination of post-induction vulvar morphology was conducted in selected studies [1,11,12,15]. Vulvar and vaginal tissue samples were meticulously prepared using a microtome, followed by staining with Hematoxylin and Eosin (H&E). This process allows for the detailed evaluation of morphological alterations, including edema, tissue thickening, and the presence of inflammatory infiltrates, which are indicative of the pathophysiological changes resulting from vulvodynia induction.

#### 2.6.2. Immunofluorescent Staining and Microscopy in LPV Animal Models

To further elucidate the underlying mechanisms of vulvodynia at the cellular level, some studies utilized advanced multiple-labeling immunohistochemistry techniques. This approach facilitated the detailed investigation of neuronal hyperinnervation and other crucial factors affecting vulvar nerve tissues. The markers used in these studies, along with their specific functions in the context of vulvodynia, are summarized in the table below (Table 5).

In addition, the studies explored markers related to neuronal neuromodulation and activity. This analysis aimed to understand the transmission of pain signals from the vulvar region to the spinal cord and to examine the extent of inflammation and the presence of immune cells within the affected tissues. The findings from these investigations provide significant insights into the molecular and cellular processes involved in vulvodynia, offering potential targets for therapeutic intervention (Table 6).

#### 2.6.3. Biochemistry and Protein Measurements in LPV Animal Models

To deepen our understanding of the cellular mechanisms contributing to the onset and persistence of vulvar pain, biochemical analyses were conducted. These analyses focused on identifying specific cell signaling pathways that play a critical role in the development of vulvodynia. The table below (Table 7) provides a comprehensive summary of the key biochemical markers that were examined, detailing their roles and how they contribute to the pathophysiology of vulvar pain. This investigation into the biochemical landscape of vulvodynia helps to pinpoint potential molecular targets for therapeutic intervention and enhances our understanding of the condition’s underlying biological processes.

#### 2.6.4. Gene Expression in LPV Animal Models

To assess gene expression alterations associated with chronic vulvar pain, real-time PCR (Polymerase Chain Reaction) was employed. This technique allowed for the precise quantification of gene expression changes in the labial tissue and spinal cord of rats afflicted with vulvodynia. Total RNA was extracted from these tissues in various studies, and the expression levels of select genes were measured and normalized against housekeeping genes to ensure accuracy (Table 8) [16,19]. Additionally, in one of the studies, concentrations of IgE, CXCL2, and IL-1β were determined using Enzyme-Linked Immunosorbent Assay (ELISA) [26]. These molecular analyses offer invaluable insights into the genetic factors contributing to vulvodynia and highlight potential biomarkers for diagnosing and monitoring the progression of the disease.

## 3. Results—The LPV Animal Models

### 3.1. LPV Animal Models Induced by Inflammation

Animal LPV models induced by inflammation have played a crucial role in vulvodynia research, highlighting the impact of inflammation and immune responses on the disorder. These models have utilized various methods to induce vulvodynia for study.

### 3.2. Candida/Zymosan for Induction of LPV Animal Model

The Candida/zymosan model was developed in response to observations that over 70% of women diagnosed with LPV have reported a history of recurrent candidiasis [23]. The subsequent preference for zymosan, as a surrogate for live yeast, in inducing vulvodynia in animal models, stems from its efficacy and convenience [13].

#### 3.2.1. First Animal Model of LPV Induced by Candida/Zymosan

In 2011, Farmer et al. [12] pioneered the induction of a vulvodynia model in mice through repeated vulvovaginal infections with C. *albicans* or zymosan. They demonstrated a long-lasting (70 days) allodynia, characterized by at least a 33% decrease in mechanical sensitivity, following three rounds of C. *albicans* infection. Similarly, allodynia was triggered by zymosan injections, although the number of injections required for chronic pain induction varied among mice (Table 9A).

#### 3.2.2. Morphological Changes in LPV Animal Models Induced by Candida/Zymosan

Following infection with C. albicans, post-scarification H&E staining revealed no edema or presence of inflammatory cells (Table 9A). Notably, there was a significant increase in the density of sympathetic and sensory nerve fibers, specifically labeled for CGRP and VMAT2 within the lamina propria, persisting for a minimum of three weeks [12] (Table 9A).

#### 3.2.3. Blocking Trial in LPV Animal Models Induced by Candida/Zymosan

The rationale behind conducting blocking trials in the context of LPV research is to establish the causative role of inducing agents in the development of LPV more definitively. By utilizing blocking agents capable of preventing LPV’s onset, these trials aim to demonstrate that the induction agents are directly responsible for initiating the pathological processes leading to LPV. This approach not only helps in confirming the etiological factors of LPV but also aids in identifying potential therapeutic targets for intervention, thereby enhancing our understanding of LPV’s pathogenesis and informing the development of effective treatment strategies.

In a blocking trial, treatment with fluconazole, a common antifungal agent used in humans [81], was administered for 7 days, starting on day 4, following each C. albicans administration to mice. This treatment resulted in a reduction of allodynia after the first two rounds of C. albicans administration but failed to produce the same effect in the third round [12]. It is noteworthy that fluconazole has not demonstrated efficacy in alleviating established provoked vulvodynia in women [82] (Table 9A).

#### 3.2.4. Fibroblast, SPM, IL-6, and PGE2 Study in LPV Animal Models Induced by Candida/Zymosan

In 2021, Falsetta et al. [13] successfully induced vulvodynia in mice, as evidenced by decreased mechanical sensitivity thresholds after up to six weekly zymosan injections into the vagina. The majority of mice developed long-lasting (91 days) allodynia after receiving four injections (Table 9A).

#### 3.2.5. Morphological Changes and Biochemistry Measurement in LPV Animal Models Induced by Candida/Zymosan

In the study by Falsetta et al. [13], H&E staining was not performed. However, signs of local inflammation were observed in the vicinity of the infection site, which was resolved soon after the last zymosan injection. Notably, the levels of Prostaglandin E2 (PGE2) peaked following the fourth zymosan injection, indicating a biochemical response to the induced inflammation (Table 9A). 

#### 3.2.6. Blocking Trial in LPV Animal Models Induced by Candida/Zymosan

In their 2021 study, Falsetta et al. [13] explored the effects of treating mice with topical Marine 1, a specialized pro-resolving mediator (SPM), on vulvodynia induced by zymosan injections. The treatment commenced one week following the last zymosan injection and consisted of daily applications for 7 weeks. By the fourth week of treatment, an increase in mechanical sensitivity thresholds was observed, nearing or surpassing baseline levels. However, changes in Prostaglandin E2 (PGE2) levels after Marine 1 treatment did not reach statistical significance (see Appendix A).

Additionally, mice treated with topical docosahexaenoic acid (DHA) twice daily for a total of 7 weeks also exhibited increased mechanical sensitivity thresholds. It is important to highlight that the study did not investigate neuroproliferation [13] (Table 9A).

#### 3.2.7. Investigation of Mast Cell Role in LPV Animal Models Induced by Candida/Zymosan

Awad-Igbaria et al., in 2022 [11], successfully induced long-lasting vulvodynia (156 days) in rats through three repeated zymosan injections. This study highlights the potential involvement of mast cells in the persistence and development of vulvodynia, offering a significant duration for the observation and analysis of the condition’s progression (Table 9A).

Following the zymosan injections, H&E staining revealed significant dermal lymphocytic infiltration with exocytosis, indicating a pronounced inflammatory response in the tissue (Table 9A). An increase in the number of activated and degranulated mast cells (MCs) was observed 20 days after the zymosan challenge, which, however, was not evident after 160 days (Table 9A). This temporal pattern suggests an initial acute inflammatory response that subsides over time.

Furthermore, there was a noted increase in neuroproliferation, alongside heightened expression of pain channels (TRPA1 and TRPV1) and neurotransmitters (NGF and glutamate), indicating the complex interplay of inflammatory cells, nerve growth factors, and pain signaling pathways in the development and persistence of vulvodynia [51] (Table 9A).

#### 3.2.8. Blocking Trial in LPV Animal Models Induced by Candida/Zymosan

Ketotifen fumarate (KF), recognized for its mast cell (MC) stabilizing, antiallergic, and antihistaminergic properties, inhibits MC degranulation and histamine release [83]. In the study by Awad-Igbaria et al. [11], pre-treatment with KF prior to each zymosan challenge, followed by daily administration for one week, led to significant improvements in allodynia. Additionally, there was a notable decrease in nerve density, nerve growth factor (NGF) levels, and the number of MCs (Table 9A). These findings underscore the critical role of mast cells in the pathogenesis of Localized Provoked Vulvodynia (LPV), further validating the therapeutic potential of targeting MC activity in managing this condition.

### 3.3. The Complete Freund’s Adjuvant (CFA)-Induced LPV Animal Model

In 2018, Sharma et al. [1] injected CFA into the distal vagina in mice. A weakness of this study is the lack of testing for the presence of vulvodynia, treatment, and prevention methods (Table 9B).

#### 3.3.1. Morphological and Immunohistochemistry Changes in LPV Animal Models Induced by CFA

Sharma et al. [1] utilized H&E staining to investigate the effects of CFA injections on the vulvar tissue. Fourteen days after injection, alterations in the lamina propria were noted, which were subsequently resolved by day 28. The CFA injections led to an increase in nerve fibers, specifically those labeled for CGRP (Calcitonin Gene-Related Peptide) and SP (Substance P), as well as fibers expressing VIP (Vasoactive Intestinal Peptide). Additionally, there was an observed increase in the number of blood vessels labeled for α-SMA (Alpha Smooth Muscle Actin) (Table 9B).

A notable rise in CD-68 positive macrophages was observed following repeated injections of CFA, persisting for at least two weeks. In contrast, no significant increase in mast cell numbers was detected (Table 9B). This differential cellular response underscores the complex immune dynamics at play following CFA-induced inflammation and highlights the specific role of macrophages in this model of vulvodynia.

In a study by Castro et al. [15], immunohistochemistry staining revealed an increase in phosphorylated extracellular signal-regulated kinase (pERK) neurons within the dorsal horn of the lumbosacral (L6-S1) spinal cord, indicating enhanced pain signal transduction. Additionally, both M1 and M2 macrophages showed increased presence, suggesting an activated immune response. Hyperinnervation was also noted in the CFA-injected mice. Despite these substantial changes at the cellular and molecular levels, morphological changes were not evident upon Hematoxylin and Eosin (H&E) staining (Table 9B). This discrepancy highlights the importance of employing multiple staining techniques to fully understand the effects of CFA injections on vulvodynia models.

In another study, by Chakrabaty et al. [14], Hematoxylin and Eosin (H&E) staining was not performed to assess morphological changes. However, signs of local inflammation were noted near the site of CFA injection after one week, indicating an immediate response to the inflammatory challenge. A notable observation was the occurrence of hyperinnervation, characterized predominantly by an increase in nerve fibers expressing Calcitonin Gene-Related Peptide (CGRP) and Glial Cell Line-Derived Neurotrophic Factor Family Receptor Alpha 2 (GFRalpha2).

Additionally, there was a significant increase in macrophages and T cells expressing renin (REN) and angiotensinogen (AGT), along with the degranulation of mast cells (MCs). This suggests an intricate interaction between the nervous and immune systems in the development of vulvodynia after CFA injection. Contrary to these findings, no changes were observed in the number of B cells or in B cells expressing AGT or REN (Table 9B). This specificity in cellular response highlights the selective involvement of certain immune cells in the pathogenesis of CFA-induced vulvodynia.

#### 3.3.2. The Study of Visceromotor Response (VMR) in LPV Animal Models Induced by CFA

Castro et al., in 2022 [15], further explored the induction of vulvodynia through the injection of Complete Freund’s Adjuvant (CFA) in mice. Following the CFA administration, the mice were monitored closely twice daily for a period of 7 days. The assessment of mechanical sensitivity was conducted using a vaginal balloon catheter, which demonstrated an increase in the visceromotor response (VMR) after CFA administration (Table 9B). This increase in VMR indicates heightened sensitivity and discomfort, underscoring the effectiveness of CFA in modeling vulvodynia-related symptoms in mice.

#### 3.3.3. The Study of Renin–Angiotensin System in LPV Animal Models Induced by CFA

In a study conducted by Chakrabaty et al. in 2017 [14], vulvodynia was induced in rats by injecting Complete Freund’s Adjuvant (CFA) into their posterior vestibule. The induction of allodynia was observed one week following the CFA injection. This timeframe highlights the rapid onset of vulvodynia symptoms following inflammatory insult in this model, further emphasizing the potential role of the renin–angiotensin system in the pathophysiology of vulvodynia (Table 9B).

#### 3.3.4. Blocking Trials in LPV Animal Models Induced by CFA

To further elucidate the role of macrophages in the development of vulvodynia, clodronate was utilized as a macrophage-depleting chemical agent. Its mechanism involves inducing apoptosis in macrophages by interfering with mitochondrial activity [84]. The administration of clodronate alongside CFA on day 0 and day 4 resulted in a reduction of the visceromotor response (VMR), normalization of phosphorylated extracellular signal-regulated kinase (pERK) activation levels to baseline, and a decrease in the numbers of both M1 and M2 macrophages. Despite these improvements, treatment with clodronate did not mitigate hyperinnervation (Table 9B). This trial underscores the potential of targeting macrophage activity in alleviating certain vulvodynia symptoms, although it also highlights the complexity of addressing nerve fiber density changes associated with the condition.

In another blocking trial, all rats received intraperitoneal (IP) injections of PD123319 difluoroacetate (PD), an antagonist of the angiotensin II type 2 (AT2) receptor, starting from day 0 and continuing daily for 7 days. This treatment was aimed at investigating the therapeutic potential of AT2 receptor inhibition in the context of CFA-induced vulvodynia.

The results showed a reduction in pain levels and a decrease in the density of CGRP (Calcitonin Gene-Related Peptide) and GFRalpha2 (Glial Cell Line-Derived Neurotrophic Factor Family Receptor Alpha 2) nerve fibers to baseline levels following PD treatment. Additionally, PD administration led to a decrease in the number of mast cells (MCs) undergoing degranulation and in T cells expressing renin (REN) and angiotensinogen (AGT). However, it did not significantly alter the number of macrophages expressing AGT or REN (Table 9B).

These findings underscore the efficacy of targeting the AT2 receptor pathway as a means to mitigate pain and nerve fiber density in vulvodynia models, as well as to modulate specific immune cell responses associated with the condition (see Appendix A).

### 3.4. Induction of LPV Animal Models by Allergy

The allergy models are founded on the hypothesis of a connection between allergic reactions and the development of vulvodynia, emphasizing the involvement of histamine, interleukin receptors, and mast cells (MCs), which have been implicated in the pathogenesis of vulvodynia [85]. These models employ various allergens, such as oxazolone (OX) and methylisothiazolinone (MI), to provoke allergic reactions. By doing so, researchers aim to study the potential mechanisms through which these allergic responses contribute to vulvodynia, exploring the role of immune system components and inflammatory mediators in the condition’s onset and progression.

### 3.5. Oxazolone (OX) for Induction of LPV Animal Models

The choice to employ oxazolone in the study of vulvodynia models is informed by its established use in creating models for various inflammatory and allergic conditions. Specifically, oxazolone has been effectively utilized to simulate allergic contact dermatitis [86], ulcerative colitis [87], and induced orbit inflammation [29]. This versatility in inducing specific types of inflammation and allergic reactions makes oxazolone a valuable tool for investigating the allergic pathophysiological mechanism of vulvodynia.

#### 3.5.1. The Study of Cytokines in LPV Animal Models Induced by OX

In their research, Martinov et al. [16] explored the effects of oxazolone (OX) in inducing vulvodynia in mice through both single and triple topical applications. The study observed peak hyperalgesia—a heightened sensitivity to pain—lasting for 24 h after a single OX application and for five days following the triple application. The hyperalgesia then subsided after two days and ten days, respectively. This investigation provides valuable insights into the temporal dynamics of pain sensitivity following allergen exposure in vulvodynia models. However, a notable limitation of this study is the relatively short duration of the observation period, which was capped at 10 days. This constraint might limit the understanding of the longer-term effects of OX-induced vulvodynia and its potential resolution or chronicity (Table 10).

#### 3.5.2. Immunohistochemistry and PCR Measurements in LPV Animal Models Induced by OX

The study conducted by Martinov et al. [16] highlighted significant changes at the cellular and molecular levels following topical OX application. Immunohistochemistry results showed an increase in the number and activity of neutrophils and eosinophils. Additionally, an increase in PGP9.5 and CGRP (Calcitonin Gene-Related Peptide) nerve fibers was observed after both single and triple OX applications, indicating neurogenic inflammation and potential nerve fiber remodeling.

Mast cell (MC) degranulation was particularly noted to increase 24 h after the single OX application, reflecting immediate hypersensitivity reactions contributing to the vulvodynia model.

Polymerase Chain Reaction (PCR) measurements revealed elevated transcripts of pro-inflammatory cytokines and chemokines, including IL-1β, TNF-α, CXCL1, CXCL2, IL-6, and IFN-γ, during the early inflammatory phase, 6–24 h after OX application. These levels were resolved after 48 h, indicating a transient inflammatory response. This dynamic expression of cytokines and chemokines after OX application underscores the complex interplay of immune responses in the development of vulvodynia (Table 10).

#### 3.5.3. Blocking Trial in LPV Animal Models Induced by OX

In an effort to block the effects of oxazolone-induced vulvodynia, a prophylactic approach was tested using mast cell stabilization. Sodium cromoglycate (SCG), known for its ability to stabilize mast cells and prevent their degranulation, was applied prior to the topical OX application. This intervention resulted in a notable decrease in allodynia at 1 and 6 h after application, demonstrating the effectiveness of SCG in the immediate aftermath of OX exposure. However, this protective effect did not persist as no significant reduction in allodynia was observed after 24 h. This suggests that while mast cell stabilization can mitigate early symptoms of vulvodynia induced by allergic reactions, its effectiveness diminishes over time (Table 10).

#### 3.5.4. The Study of T Cells in LPV Animal Models Induced by OX

In their 2017 study, Landry et al. [19] explored the role of T cells in vulvodynia by applying oxazolone (Ox) topically to the vulva of mice, totaling ten applications. Remarkably, just one day following the final OX application, significant allodynia was observed, indicating a heightened sensitivity to pain that closely mimics the condition in humans. Allodynia persisted for at least 21 days, demonstrating the prolonged impact of the allergen on inducing pain sensitivity. However, it is noteworthy that allodynia returned to baseline levels after 42 days, suggesting a temporal limitation to the allergen’s effect or possibly the activation of mechanisms leading to the resolution of the condition. This study provides valuable insights into the persistence and resolution phases of vulvodynia induced by an allergic reaction, highlighting the potential involvement of T cells in the condition’s pathophysiology (Table 10).

#### 3.5.5. Morphology, Immunohistochemistry, and PCR in LPV Animal Models Induced by OX

In the study conducted by Landry et al. [19]., while overt signs of tissue inflammation were not evident upon visual inspection—partly due to examinations occurring after 21 days—Hematoxylin and Eosin (H&E) staining was not performed, which could have provided detailed insights into microscopic changes. Despite the lack of visible inflammation, significant changes were observed at the molecular and cellular levels:An increase in Nerve Growth Factor (NGF) levels, Calcitonin Gene-Related Peptide (CGRP) nerve fibers, mast cell (MC) numbers, and histamine levels were documented. Notably, NGF levels began to decrease gradually and disappeared by day 21, while the latter three parameters only returned to baseline on day 42.Additionally, significant elevations in total Immunoglobulin E (IgE), interleukins (IL-3 and IL-6 mRNA), cellular adhesion molecule 1 (Camd1), Tumor Necrosis Factor-alpha (TNF-alpha), and chemokine (C-X-C motif) ligand 2 (CXCL2) were observed one day after the final OX application, indicating an acute inflammatory and immune response. All these markers returned to baseline levels by day 21.

These findings highlight the complex interplay between nerve fiber activity, mast cell activation, and the release of histamine and other cytokines following allergen exposure. The gradual normalization of these parameters suggests a resolution phase that aligns with the alleviation of allodynia symptoms. The delayed return to baseline levels for certain markers underscores the prolonged impact of OX application on inducing and sustaining vulvodynia-related changes (Table 10).

#### 3.5.6. T Cell Profile in LPV Animal Models Induced by OX

Regulatory T cells [88] and IFN-gamma have been linked to MC accumulation in vulvodynia models [89,90]. Furthermore, Landry et al. (2017) [19] observed a distinct T cell profile characterized by CD4, CD25, and FOXP3, along with a significant increase in IFN-gamma, potentially facilitating mast cell recruitment (Table 10).

#### 3.5.7. Blocking Trial in LPV Animal Models Induced by OX

Administered daily between days 5–8 after the 10th OX application, C48/80—an MC degranulation inhibitor—significantly reduced MC density and CGRP nerve fibers, while enhancing pain thresholds. These effects were observed even 21 days after treatment (Table 10).

### 3.6. Models Using Methylisothiazolinone (MI) for Induction of LPV Animal Models

Recent research indicates that exposure to environmental toxins, including biocide preservatives, could lead to vulvodynia [91].

#### 3.6.1. Spinal Cord Transcript Study in LPV Animal Models Induced by MI

Arriaga-Gomez et al. (2019a) [18] triggered vulvodynia in mice using 10 daily topical MI applications to the vaginal canal, with allodynia persisting for at least 14 days (Table 10).

#### 3.6.2. Immunohistochemistry and IgE Levels in LPV Animal Models Induced by MI

After 10 daily topical MI applications, an increase in circulating IgE levels, MC numbers, and activated eosinophils was noted one day after treatment, normalizing after at least 3 weeks. No rise in neutrophil count was observed (Table 10).

#### 3.6.3. mRNA Extraction from Vaginal Canal and Spinal Cord in LPV Animal Models Induced by MI

mRNA transcripts from the vaginal canal and spinal cord were extracted, revealing IFN-γ and IL-6 mRNA from the vaginal canal and IL-1β and IL-6 mRNA from the spinal cord detected one day after the 10th application. All transcripts normalized to baseline within 7 days at both sites (Table 10).

#### 3.6.4. T Cell Study with MI Application in LPV Animal Models Induced by MI

Kline et al. (2020) [17] reported vulvodynia lasting up to 70 days after 10 daily MI applications to labial skin (Table 10).

#### 3.6.5. Immunohistochemistry, T Cell Profile, and Transcripts in LPV Animal Models Induced by MI

Increased MC density and IgE levels persisted for at least 49 days following the 10th topical MI application. Activated eosinophils and neutrophils also increased, with labial skin samples showing accumulations of CD4+ and CD8+ T cells stained for CD44+, CD25+, and CD103+. These increases were observed one day after the tenth challenge but returned to normal by day 21 (Table 9). Repeated MI applications triggered the overexpression of pro-inflammatory cytokines Cxcl2, IL-1β, IL-6, IFN-γ, and Tbx21 mRNA transcripts (Table 10).

#### 3.6.6. Blocking Trials in LPV Animal Models Induced by MI

Δ-9-tetrahydrocannabinol (THC), a phytocannabinoid [92], modulates MC number and activity via Cannabinoid receptor-1 (CB1R) and neurogenic inflammation [93,94]. THC administration, both as treatment one day after MI application and as prevention 12 h before each MI application, normalized sensitivity thresholds and MC numbers within a week after the final MI administration (Table 10).

In Kline’s study, daily Imatinib treatment for 6 days following the 10th MI challenge successfully reduced mast cell density and allodynia to baseline levels (Table 10).

### 3.7. Induction of LPV Animal Models by Hormonal Imbalance

Hormonal imbalances, especially concerning progesterone and estrogen, are implicated in vulvodynia’s etiology. Studies have shown that the use of oral contraceptives, particularly those with high progestogenic and androgenic activity, correlates with heightened sensitivity of the vestibular mucosa and an increased risk of LPV [95,96]. The highest risk is associated with early initiation of contraceptives containing progesterone [97].

### 3.8. Hormone Study in LPV Animal Models Induced by Hormonal Imbalance

Liao and Smith (2014) [9] examined genital hyperinnervation in rats starting on postnatal day 20, administering daily progesterone for seven days. A subgroup was sacrificed on day 28, and on day 43, the remaining subjects underwent bilateral ovariectomy (OVX). They were then implanted with a pellet containing 17β-estradiol (E2), elevating serum E2 to levels akin to term pregnancy [64], or a placebo (Table 11).

### 3.9. Immunohistochemistry in LPV Animal Models Induced by Hormonal Imbalance

By day 28, there was an observed increase in PGP9.5 vaginal nerve fiber density. Additionally, increases in TH- and CGRP- (but not VAChT-) labeled nerve fibers were noted. By day 50, progesterone-treated OVX rats exhibited elevated densities of TH and CGRP nerve fibers (Table 11).

This streamlined summary captures the essential observations regarding nerve fiber density changes over the study period.

### 3.10. Blocking Trial in LPV Animal Models Induced by Hormonal Imbalance

Estradiol 17-β (E2) induces neurogenic inflammation via synergy with mediators such as NGF and the activation of vanilloid receptors [15,53]. A 7-day E2 treatment decreased nerve density in mice previously administered with progesterone (Table 11).

## 4. Discussion

This review offers an exhaustive overview of the methodologies utilized for inducing LPV in animal models, encompassing induction methods, pain measurement techniques, morphological evaluations, and immune cell and cytokine profiling. Initial analyses indicate LPV’s multifactorial nature, potentially driven by inflammatory responses, allergies, and immunological and hormonal influences.

### 4.1. Application of Findings to Humans

There are limitations to generalizing these factors to humans. Differences in physiology, anatomy, and pain perception between animals and humans can significantly impact the applicability of these findings to human patients. 

In addition, the data available in the literature on animal models of vulvodynia are not robust and, therefore, cannot be used for analyzing the quality and reproducibility of the data. 

### 4.2. Induction of Vulvodynia

Successful induction of allodynia followed challenges with C. albicans, zymosan, CFA, oxazolone, and MI through injections or topical applications [11,12,13,17,19]. Each method initiated LPV via unique mechanisms, suggesting disease development results from several processes.

The progesterone challenge and Sharma et al. (2018)’s [1] study did not assess pain, preventing classification as LPV animal models.

### 4.3. Site of Injection

There is no unified approach regarding injection sites, with studies employing vulvar, labial injections, or vaginal applications. This variation underscores the need for standardized methodologies to enhance comparability. Injections can cause an acute local painful reaction, while topical application may be disadvantaged because the animal may lick the material applied.

### 4.4. Study Period

Distinguishing between chronic and acute pain models is critical as short-term allodynia might result from local tissue damage rather than LPV. Consequently, studies with brief follow-up periods [14,15,16,18] pose interpretative challenges in representing LPV in animal models.

### 4.5. Selection of the Animal and Ethical Considerations

The use of animal models in research needs to be conducted according to universal ethical concerns. All the papers discussed in this review mentioned that they followed the National Institutes of Health guidelines. The female Sprague Dawley rat is favored in numerous studies [9,11,14] and was chosen by our research group [11] due to its larger size, facilitating genital organ studies. The rationale for species selection remains unspecified in other studies. The use of other animals as models has not yet been described, although the study of models with other animals may be essential for the validity of the research findings of the current review.

### 4.6. H&E Assessment

Histopathological analysis was performed in four studies [1,11,12,15], identifying transient edema and erythema in half of these [1,11]. In studies that did not observe edema, morphological evaluations were conducted after two weeks, potentially after the resolution of initial alterations. This timing might reflect the progression in women, where allodynia persists beyond the acute inflammatory stage [98]. The lack of clarity on tissue sample origins in most studies merits attention for clarity and specificity.

### 4.7. Hyperinnervation and Pain Channels

Biopsies from women with LPV consistently show hyperinnervation as a defining pathological characteristic of LPV [99,100,101]. Research across all animal models has identified an increase in both sensory and sympathetic nerve fibers, with CGRP-labeled nerve fibers emerging as the dominant subpopulation across models. Yet, biopsies from women present mixed findings on CGRP nerve fiber changes [102,103]. Other identified nerve fiber subtypes include SP, VIP, and GFRα1 following CFA challenges, while TH nerve fibers were specifically noted after progesterone challenges.

Pain channels identified in biopsies from women with LPV [102,103] have also been documented in animal models. A study [11] showed the overexpression of TRPV1 and TRPA1 pain channels in the sensory neurons of the vulva in LPV-affected animals. These findings suggest that chronic pain in LPV may be mediated by neuronal hyperinnervation, highlighting the importance of neuromodulation in managing vulvar nerves and pain pathways.

### 4.8. Inflammatory Response in LPV Animal Models

All animal model studies have focused on inflammation as a critical element in allodynia development, generally revealing a biphasic inflammation pattern: an early transient response followed by chronic inflammation.

### 4.9. Early Post-Induction Tissue Response

The initial phase after zymosan and oxazolone induction saw increases in NGF and glutamate [11,19], both linked to hyperinnervation and TRPV1 activation [104,105,106]. Additionally, early overexpression of pro-inflammatory cytokines (e.g., TNF-α, IL-6, and CCL1) was noted in oxazolone and MI-challenged LPV models.

### 4.10. Immune Cells

Increased immune cell densities in vulvar vestibular tissue of LPV mimic patterns seen in other inflammatory and neuropathic pain states [107]. The exact role of these immune cells in LPV pathogenesis remains unclear, necessitating further study. Treatments focusing on these immune responses, such as clodronate and PD, have shown potential for altering pain pathways and neuronal activation, suggesting new avenues for therapy.

### 4.11. Study of Mast Cells (MCs) in LPV

Investigations into MCs after challenges with zymosan, CFA, oxazolone (Ox), and methylisothiazolinone (MI) consistently showed an increased MC count, except for one outlier study [1] (Table 9B). The rise in MCs, particularly noted after zymosan and OX challenges, supports the theory that MC-derived NGF plays a pivotal role in enhancing synaptic transmission from peripheral sensory neurons, potentially leading to LPV [108,109].

To further explore MCs’ contribution to LPV development, two MC stabilizers (KF and SCG) and the MC degranulator inhibitor C48/80 were tested across various animal models. These interventions either modulated or prevented LPV onset, underscoring MCs’ critical role in disease pathology (Table 9A).

### 4.12. Clodronate Treatment

Clodronate’s depleting effect on M1 and M2 macrophages, without affecting hyperinnervation but normalizing pERK activation, suggests it may inhibit neuronal activation and nociceptor signal amplification. This proposed link between macrophage depletion and neuronal activation warrants further investigation.

### 4.13. Considerations for Prophylaxis

While prophylaxis is an actionable strategy in animal models, its feasibility in women remains uncertain, especially as most LPV patients seek medical care well after symptoms begin. This gap highlights the need for research into early intervention and prevention strategies in human LPV management.

### 4.14. PD as a Therapeutic Strategy

PD, targeting AT2 blockade, was investigated in animal models (Table 9B). The capacity of AT2 blockade to reduce nerve growth suggests that angiotensin II (AGII) is significant in promoting hyperinnervation. This finding underscores AGII’s potential role in the pathogenesis of LPV and suggests AT2 as a novel target for intervention.

### 4.15. Marine 1—Specialized Pro-Resolving Mediators

Marine 1 application was observed to alleviate allodynia without affecting PGE2 levels, contrasting with previous in vitro findings in vestibular fibroblasts [24,110,111,112]. The analgesic mechanism of specialized pro-resolving mediators (SPMs) in this context remains unclear, highlighting an area for further research to uncover the underlying processes.

### 4.16. THC Treatment

THC, a phytocannabinoid, showed effectiveness in reducing mast cell accumulation and allodynia in an LPV animal model, as detailed in Table 10. This outcome supports the potential therapeutic use of cannabinoids in managing LPV by modulating immune responses and pain perception.

### 4.17. Estradiol (E2) Treatment

E2 treatment after progesterone challenge led to decreased hyperinnervation (Table 11). However, translating hormonal interventions from animal models to humans is complex given the difficulty in replicating the intricate endogenous hormonal dynamics experienced throughout maturation and the menstrual cycle. Notably, pain responses were not evaluated in the progesterone-challenged models, presenting a limitation. The findings suggest an interplay between progesterone and E2, with E2 potentially reducing innervation density, indicating that early progesterone exposure might lead to permanent increases in innervation.

These insights into various therapeutic approaches offer promising directions for LPV treatment, each with unique mechanisms of action. Further investigation into these treatments will help clarify their efficacy and potential application in human LPV management. Such studies may include research into novel treatments with cannabinoids, various agents blocking each step of the mechanism of the development of vulvodynia, and the involvement of the microbiome. Further studies can be conducted on the inhibition of MC activity and research on the involvement of the central nervous system in the development of vulvodynia and its treatment.

In addition, the involvement of other factors, including hormonal, immunological, and genetic issues, in the development of LPV should also be studied in animals.

## 5. Conclusions

The deployment of animal models in LPV research has been instrumental in deepening our comprehension of this condition’s pathogenesis. The bulk of studies utilizing these models have showcased vestibular allodynia, neuroproliferation, and an enriched variety in nerve fiber density. Notably, one investigation highlighted enhanced spinal cord neuron activation, and another identified increased pain channel expression in LPV-induced animals. Additional findings across some studies include nociceptor sensitization, abnormal pro-inflammatory signaling, and a marked presence of inflammatory cells, notably those expressing renin–angiotensin system proteins. Furthermore, elevated counts of immune cells like macrophages and mast cells, alongside a temporary surge in inflammatory cytokines, were documented.

This review meticulously covers the assortment of animal models adopted, the induction techniques applied, and the wide range of variables explored, both at the tissue and behavioral levels. Consequently, it lays a solid groundwork for future inquiries in this domain, encouraging further investigation into LPV’s underlying mechanisms and potential treatment avenues. This publication can be regarded as a starting point for subsequent analyses of animal models, which are essential to understanding the pathogenesis of this complex multifactorial disease.

## Figures and Tables

**Table 1 ijms-25-04261-t001:** Strains of Rats and Mice Used in Studies.

Species	Strain	Age Range (Weeks)	Study References	Type
Rat	Sprague Dawley	10	[9,11,14]	Outbred
Mouse	CD-1	8–10	[12]	Outbred
Mouse	ND4	6–12	[16,17,18,19]	Outbred
Mouse	C57BL/6	6–12	[13,15]	Inbred

**Table 2 ijms-25-04261-t002:** Induction Method Substance Properties.

Substance	Origin	Description	Reference
*Candida albicans (C. albicans)*	Yeast	Induces local tissue inflammation	[23]
Zymosan	Produced from a yeast cell wall	A highly stimulatory protein–carbohydrate complex. Provokes inflammation and hyperalgesia through activation of TRP channels and excitatory effect mediated by glutamate.	[24,25]
Complete Freud adjuvant (CFA)	Compound containing heat-inactivated *Mycobacterium butyricum* in oil	Induces dose-related inflammatory response. Maximal effect is observed 6–8 h after administration.	[26,27,28]
Oxazolone	Hapten, a chemical compound	Stimulates a type 4 hypersensitivity cascade that triggers the adaptive immune response. The response is mediated by a mast cell immunoglobulin IgE.	[29,30,31,32]
Methylisothiazolinone (MI)	Potent antimicrobial sensitizer agent	Found in a wide range of cleaning and cosmetics products, soap, and the paint industry. It is estimated that 2–10% of the population is MI-sensitive by patch testing.	[18,33,34]
Progesterone	Hormone	Increases neurotrophic factors in the female reproductive tract and has been shown to boost axon sprouting and myelination.	[35,36]

**Table 3 ijms-25-04261-t003:** Induction Methods for Vulvodynia in Animal Models.

Study	Year	Induction Method	Substance Used	Volume/Dose	Application Site
Farmer et al. [12]	2012	Infection and Injection	*Candida albicans (C. albicans)* strain SC5314, Zymosan	0.1 mg	Posterior vulva
Falsetta et al. [13]	2021	Injection	Zymosan	0.1 mg	Midline posterior vulva
Awad-Igbaria et al. [11]	2022	Injection	Zymosan	0.1 mg	Both sides of the vulva
Castro et al. [15]	2022	Injection	Complete Freud’s adjuvant (CFA)	5 µL	Vaginal wall at the introitus
Sharma et al. [1]	2018	Injection	CFA	5 µL	Wall of the distal vagina
Chakrabarty et al. [14]	2018	Injection	CFA	30 μL	Posterior perivaginal vestibular tissue
Martinov et al. [16]	2013	Topical Application	1% Ox	40 μL	Labia
Landry et al. [19]	2017	Topical Application	1% Ox	40 μL	Genital skin, including labia
Arriaga-Gomez et al. [18]	2019	Topical Application	0.5% Methylisothiazolinone (MI)	20 μL	Labia
Kline et al. [17]	2020	Topical Application	0.5% MI	40 μL	Labia
Liao and Smith et al. [9]	2014	Subcutaneous Injection	Progesterone	20 mg/kg	Not specified

**Table 4 ijms-25-04261-t004:** Mechanical Sensitivity Testing Methods in Animal Models of Vulvodynia.

Technique	Method	Description	Reference
Manual	Up-Down Psychophysical	Utilization of von Frey filaments to measure mechanical sensitivity. Pressure is increased until a behavioral response is observed.	[15]
Electronic	Von Frey Device	An electronic device that increases the pressure, until the subject steps or jumps. The pressure is recorded to measure mechanical sensitivity precisely.	[15]
Unique	Balloon Catheter	Used by Castro et al., 2022 [15], to distend the vaginal canal and assess visceromotor response (VMR), indicating nociceptive involuntary abdominal muscle contraction.	[37]

**Table 5 ijms-25-04261-t005:** Markers Identified in Vaginal Sections.

Marker	Description	Function	References
Protein gene product 9.5 (PGP9.5)	Pan axonal marker	Peripheral nerve fiber identification	[38]
Calcitonin gene-related peptide (CGRP)	Amino acid peptide	Found in sensory fibers; usually found in perivascular localization and has sensory and efferent functions	[39]
Vesicular monoamine transporter 2 (VMAT2)	Presynaptic protein	Regulates dopamine and monoamine release into synapse	[40]
Substance P (SP)	Neuropeptide	Overexpressed in nociception and chronic pain; histamine release from mast cells	[41,42]
Neuropeptide Y (NPY)	Peptide	Present in CNS *, PNS *, peripheral tissues, and blood vessels	[43,44]
Vasoactive intestinal polypeptide (VIP)	Neuropeptide	Vasomotor regulation; increases vaginal blood flow, lubrication, and anti-inflammatory effects in the vagina	[45]
Tyrosine hydroxylase (TH)	Protein	Sympathetic neuronal marker; suggested role in pain development	[46,47]
Vesicular acetylcholine transporter—(VAChT)	Integral membrane protein	Labels cholinergic parasympathetic neurons; acetylcholine release	[48,49]
GDNF-family receptor-α2 (GFRα2)	Receptor	Mechanosensory neurons; extends unmyelinated C-fibers to targets	[50]

* Central (CNS) and peripheral (PNS) nervous systems.

**Table 6 ijms-25-04261-t006:** Markers for Neuronal Neuromodulation and Immune Cell Presence.

Marker	Description	Function	References
Transient receptor potential vanilloid 1 (TRPV1) and ankyrin 1 (TRPA1)	Capsaicin/heat-activated and cold/mechanical stimuli-activated channels	Transduce noxious thermal and mechanical stimuli	[51]
Phosphorylated extracellular signal-regulated kinase (pERK)	Protein kinase	Activated by peripheral stimuli; involved in chronic pain development	[52,53]
α-smooth muscle actin (αSMA)	Actin isoform	Involved in inflammation and wound healing; labels smooth muscle blood vessels	[1,54,55]
CD-68	Glycoprotein	Marker for macrophages; role in inflammation and resolution	[56]
CD11c	Hematopoietic cell protein	M1 macrophage marker; proinflammatory effects	[57,58]
CD206	Macrophage marker	M2 macrophage marker; downregulates inflammation	[59,60]
CD79	Protein	Expressed on B cells and neoplastic B cells	[61]
T cell receptor (TCR)α/β	Receptor	Identified in T cells; leads to T cell activation	[62]
CD4, CD8, CD25, FOXP3, CD103, and CD44	T cell proteins	Classify T cell subtypes; involved in immune cell activity enhancement	[19,63]
Myeloperoxidase and eosinophil peroxidase	Enzymes	Measure activation of neutrophils and eosinophils, respectively	[64]
Chymase and tryptase	Enzymes in mast cells	Involved in inflammation and tissue remodeling	[65]
Renin (REN) and angiotensinogen (AGT)	Renin and angiotensinogen	Part of the renin–angiotensin system; involved in nerve growth and hypersensitivity	[66]

**Table 7 ijms-25-04261-t007:** Biochemical Markers and Their Roles in Vulvar Pain.

Biochemical Marker	Description	Role in Vulvar Pain	References
Glutamate	Major excitatory neurotransmitter in the CNS	Contributes to spontaneous pain and mechanical hypersensitivity through sensitization of TRP channels like TRPV1 and TRPA1	[67]
Nerve growth factor (NGF)	Promotes neuronal growth	Sensitizes nociceptors, inducing pain development; immediate pain response observed with NGF injection in skin sensitization studies	[68,69,70]
Prostaglandin E2 (PGE2)	Lipid mediator released by various cell types	Impacts pain signaling, peripheral and central sensitization, tissue swelling, and vascular permeability	[71,72,73]
Interleukin-6 (IL-6)	Pro-inflammatory cytokine produced by various cells	Contributes to allodynia and hyperalgesia in inflamed tissues	[74,75]

**Table 8 ijms-25-04261-t008:** Gene Expression Changes in Vulvar Pain Models.

Gene	Description	Role in Vulvar Pain	Study References
Interleukin-1β (IL-1β)	Cytokine	Contributes to hyperinnervation in vulvodynia	[18,76]
Tumor necrosis factor-α (TNF-α)	Cytokine	Increased in biopsies from vulvodynia patients	[77]
IL-6	Cytokine	Associated with inflammation and pain signaling	[77]
Interleukin-3 (IL-3)	Cytokine	Elevated in vulvodynia cases	[77]
Chemokine ligand 2 (CXCL2) and chemokine ligand 1 (CXCL1)	Chemokines	Attract immune cells; involved in inflammation	[78]
Interferon-γ (IFN-γ)	Cytokine	Induces mast cell remodeling	[78]
T-box21 (Tbx21)	Transcription factor	Activates cytokines; involved in immune response and production of IFN-γ	[79]
Cell adhesion molecule 1 (Camd1)	Cell adhesion molecule	Mediates adhesion and communication between mast cells and sympathetic nerves	[80]

**Table 9 ijms-25-04261-t009:** (**A**): Summary of Inflammatory Candida/Zymosan Models in Vulvodynia Studies. (**B**): Summary of Inflammatory Complete Freud Adjuvant (CFA) Models in Vulvodynia Studies.

(A)			
Study	Study Period	Induction Method	Findings and Observations
Farmer et al. (2011) [12]	70 days	Candida/zymosan model	-Repeated vulvovaginal infection with *C. albicans* or zymosan induced long-lasting allodynia. -No morphological changes were observed. -Significant increase in nerve fiber density (CGRP and VMAT). -Fluconazole reduced allodynia after two *C. albicans* challenges but not after the third.
Falsetta et al. (2021) [13]	91 days	Zymosan injection model	-Weekly zymosan injections induced long-lasting allodynia. -Inflammation signs were observed but no H&E staining was conducted. -PGE2 levels increased after zymosan. -Treatment with marine 1 and DHA decreased allodynia; treatment with marine 1 did not reduce PGE levels.
Awad-Igbaria et al. (2022) [11]	172 days	Zymosan injection in rats	-Long-lasting thermal and mechanical vulvar allodynia after zymosan. -Increase in MCs and neuroproliferation and increase in expression of pain channels TRPA1 and TRPV1. -KF pretreatment decreased allodynia, MCs, nerve fiber density, and NGF levels.
**(B)**			
**Study**	**Study Period**	**Induction Method**	**Findings and Observations**
Sharma et al. (2018) [1]	28 days	CFA	-Vulvodynia was not proven. -Increased lamina propria thickening and hyperinnervation labeled for CGRP, SP, and VIP nerve fibers -Increased vascular proliferation vessels labeled for αSMA -Increased CD68 macrophages but no increase in MCs.
Castro et al. (2022) [15]	7 days	CFA	-CFA injection induced allodynia for a minimum of one week. -Increased VMR after CFA injection. -No morphological changes. -Increased hyperinnervation and M1-M2 macrophages and increased pERK neurons in the dorsal horn. -Treatment with clodronate reduced VMR and pERK activation to baseline levels and decreased M1-M2 macrophages but not hyperinnervation.
Chakrabarty et al. (2017) [14]	7 days	CFA	-Short-term allodynia was observed. -Hyperinnervation: mainly fibers labeled for CGRP and GFRalpha2. -Increase in macrophages, T cells expressing REN and AGT, and the degranulation of MCs. No change in B cells. -PD reduced pain, and the percentage of total cells expressing REN and AGT was reduced. Decreased degranulated MCs and T cells expressing REN and AGT. -PD normalized nerve fibers to baseline levels and did not change macrophage number.

**Table 10 ijms-25-04261-t010:** Summary of Allergy Models in Vulvodynia Studies.

Study	Study Period	Induction Method	Findings and Observations
Martinov et al. (2013) [16]	10 days	Oxazolone	-Vulvodynia was induced and lasted for a minimum of 10 days. -Increased neutrophils and eosinophils number and activity and increased PGP9.5 and CGPR nerve fibers. -Increased MCs one day after one OX topical challenge. -Increased transcripts of IL-1b, TNF-a, CXCL 1, CXCL 2, IL-6, and IFN-gamma were resolved 48 h later. -Pretreatment with SCG decreased allodynia 1–6 h after administration but not after 24 h.
Landry et al. (2017) [19]	42 days	Oxazolone	-A total of 10 topical applications induced vulvodynia for at least 21 days. -Increase in NGF, CGRP nerve fibers, MCs number, and histamine levels. -IgE, Il-3, Il-6 mRNA, Camd1, TNF-alpha, and CXCL2 increased significantly. -Increase in T cells positive for CD4, CD25, and FOXP3; increase in IFN-gamma. -MC degranulator C48/80 reduced the density of MCs and CGRP nerve fibers and increased pain thresholds.
Arriaga-Gomez et al. (2019) [18]	27 days	MI	-Allodynia lasted for a minimum of 14 days after 10 MI challenges. -Increase in circulating IgE levels, MC number, and activated eosinophils. -Transcripts of IFN-γ and IL-6 from the vaginal canal and transcripts of IL-1β and IL-6 from the spinal cord were detected one day after the 10th application. -Treatment with THC normalized allodynia and MC number. -Prevention treatment with THC reduced allodynia to less than 33% threshold and decreased MC number.
Kline et al. (2020) [17]	70 days	MI	-Long-lasting vulvodynia up to 70 days following 10 daily topical MI- challenges. -Increased MC density and IgE levels were observed for a minimum of 49 days. -Increase in activated eosinophils, neutrophils, and labial CD4+ and CD8+ T cells accumulation stained for CD44+, CD25+, and CD+103 one day after the last MI challenge. -Overexpression of pro-inflammatory cytokines Cxcl2, IL-1β, IL-6, IFN-γ, and Tbx21 mRNA transcripts. -Treatment with imatinib reduced mast cell density and allodynia to baseline levels.

**Table 11 ijms-25-04261-t011:** Summary of Hormonal Models in Vulvodynia Studies.

Study	Study Period	Induction Method	Findings and Observations
Liao and Smith (2014) [9]	30 days	Progesterone	-Increases in PGP9.5 vaginal nerve fibers and TH- and CGRP- but not VAChT-labeled nerve fibers were observed on day 28. -At Day 50, OVX rats showed increased TH and CGRP nerve fiber density. -Treatment with E2 for 7 days reduced the density of nerves but not to baseline levels.

## Data Availability

The original contributions presented in the study are included in the article and Appendix A, further inquiries can be directed to the corresponding author.

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
