# Peer review of "Exploring Localized Provoked Vulvodynia: Insights from Animal Model Research"

_ijms, 2024, doi:10.3390/ijms25084261_

Round 1

Reviewer 1 Report

Comments and Suggestions for Authors

This review is structured to bring together all animal models previously described. The analysis carefully divides the animal models in three categories, extensively  explaining   all previous literature 

The whole process is well conducted and scientific data is analysed in detail 

it can be regarded as a starting point for subsequent analyses of animal modesl, which are essential to understand tha pathogenesis of a complex multifactorial disease 

Author Response

We thank the reviewers for the time and effort dedicated to reviewing our manuscript. We have carefully considered each of your suggestions and made the necessary revisions. All changes were marked and highlighted in yellow.

Reviewer 1:

This review is structured to bring together all animal models previously described. The analysis carefully divides the animal models in three categories, extensively  explaining   all previous literature 

The whole process is well conducted and scientific data is analyzed in detail 

it can be regarded as a starting point for subsequent analyses of animal models, which are essential to understand the pathogenesis of a complex multifactorial disease.

Reply:

Thank you for your comment. We added the last sentence to the manuscript - in the conclusion section:

This publication can be regarded as a starting point for subsequent analyses of animal models, which are essential to understand the pathogenesis of this complex multifactorial disease.

Reviewer 2 Report

Comments and Suggestions for Authors

The article "Exploring Localized Provoked Vulvodynia: Insights from Animal Model Research" by Yara Nakhleh-Francis et al. provides a comprehensive review of animal models used to study the pathogenesis of provoked vulvodynia. While the article offers valuable insights, there are a few limitations and areas for improvement:

1. Generalization of Findings: The article extensively discusses various animal models but does not sufficiently address the limitations of generalizing these findings to humans. Differences in physiology, anatomy, and pain perception between animals and humans can significantly impact the applicability of these findings to human patients.

2. Specificity of Animal Models: The review outlines different animal models but lacks a critical evaluation of the specificity of these models to provoked vulvodynia. The use of models that closely mimic the human condition is essential for the validity of the research findings.

3. Methodological Limitations: The article mentions various methods for inducing vulvodynia in animal models but does not thoroughly discuss the methodological limitations. For instance, the variability in induction methods, such as injections or topical applications, and their impact on the study outcomes is not thoroughly discussed.

4. Short Duration of Observation Periods: Some studies mentioned in the article have relatively short observation periods, which may not capture the chronic nature of vulvodynia. Longer-term studies are necessary to understand the persistence of symptoms and the long-term effects of potential treatments.

5. Lack of Discussion on Treatment Efficacy: While the article mentions blocking trials and treatments used in animal models, there is a lack of detailed discussion on the efficacy of these treatments. A critical evaluation of the success rates and potential side effects of these treatments would provide a more comprehensive understanding of their applicability.

6. Ethical Considerations: The use of animal models in research raises ethical concerns, which the article does not address. A discussion on the ethical implications of using animals for vulvodynia research and the efforts to minimize animal suffering would have been beneficial.

7. Future Research Directions: Although the article concludes with promising directions for future research, it does not provide a detailed roadmap for addressing the gaps identified in the current literature. Specific recommendations for future studies, including the development of more human-like models and the exploration of novel therapeutic targets, would enhance the article's contribution to the field.

8. Lack of Interdisciplinary Perspectives: The review primarily focuses on biological and physiological aspects of vulvodynia without integrating findings from interdisciplinary research, such as psychological, social, and environmental factors that could influence the condition.

9. Inadequate Consideration of Sex and Gender Differences: The article does not address how sex and gender differences might influence the pathogenesis of vulvodynia or the response to treatments in animal models, which is crucial for developing effective and personalized interventions.

10. Insufficient Analysis of Data Quality and Reproducibility: The review does not critically assess the quality and reproducibility of the data from the studies it discusses. Given the challenges of translational research from animal models to human conditions, an analysis of data robustness and reproducibility would strengthen the review's conclusions.

11. PRISMA and QUADAS-2 guidelines. The review does not mention the accepted guidelines helpful to conduct a review article.

In summary, while the article by Yara Nakhleh-Francis et al. contributes valuable information on the use of animal models in the study of provoked vulvodynia, addressing the aforementioned shortcomings would strengthen the research's impact and relevance to human clinical practice.

Comments on the Quality of English Language

minor

Author Response

We thank the reviewers for the time and effort dedicated to reviewing our manuscript. We have carefully considered each of your suggestions and made the necessary revisions. All changes were marked and highlighted in yellow.

Reviewer 2:

The article "Exploring Localized Provoked Vulvodynia: Insights from Animal Model Research" by Yara Nakhleh-Francis et al. provides a comprehensive review of animal models used to study the pathogenesis of provoked vulvodynia. While the article offers valuable insights, there are a few limitations and areas for improvement:

  1. Generalization of Findings: The article extensively discusses various animal models but does not sufficiently address the limitations of generalizing these findings to humans. Differences in physiology, anatomy, and pain perception between animals and humans can significantly impact the applicability of these findings to human patients.

Reply:

Thank you for your comment. We added the following section to the manuscript discussion: 4.1 application of findings to human

"There are limitations of generalizing these factors to humans. Differences in physiology, anatomy, and pain perception between animals and humans can significantly impact the applicability of these findings to human patients."

  1. Specificity of Animal Models: The review outlines different animal models but lacks a critical evaluation of the specificity of these models to provoked vulvodynia. The use of models that closely mimic the human condition is essential for the validity of the research findings.

Reply:

Thank you for your comment. In the section 4.5- selection of the animal and ethical considerations in the discussion, there is already a discussion of the model that has been used in most studies. In the end of that section, we added the following paragraph:

The use of other animals as models has not yet been described. Although the use of models with other animals may be essential for the validity of the research findings of the current review.

  1. Methodological Limitations: The article mentions various methods for inducing vulvodynia in animal models but does not thoroughly discuss the methodological limitations. For instance, the variability in induction methods, such as injections or topical applications, and their impact on the study outcomes is not thoroughly discussed.

Reply:

Thank you for your comment. The manuscript is actually a review, describing all the methodologies that have been used in the studies on vulvodynia. There we mention that there is no unified approach. With each mythology that is described in our manuscript we pointed out the limitations inherited in. as for the induction methods each method may have it own advantages and disadvantages there for all methods were described.

We added the following paragraph to the discussion section 4.3- site of injection:

Injections can cause an acute local painful reaction, while topical application may be disadvantaged because the animal may lick the material applied.

  1. Short Duration of Observation Periods: Some studies mentioned in the article have relatively short observation periods, which may not capture the chronic nature of vulvodynia. Longer-term studies are necessary to understand the persistence of symptoms and the long-term effects of potential treatments.

Reply:

Thank you for your comment the disadvantage in short follow up period is already mentioned in the discussion section 4.4- study period.

  1. Lack of Discussion on Treatment Efficacy: While the article mentions blocking trials and treatments used in animal models, there is a lack of detailed discussion on the efficacy of these treatments. A critical evaluation of the success rates and potential side effects of these treatments would provide a more comprehensive understanding of their applicability.

Reply:

Thank you for your comment. In most studies the success rate or the side effects of the treatments have not been described or analyzed. The reason is because these studies of animal models in vulvodynia concentrated on the description of methodologies.

Nevertheless, in the tables of our manuscript the issues of outcome were presented where available.

  1. Ethical Considerations: The use of animal models in research raises ethical concerns, which the article does not address. A discussion on the ethical implications of using animals for vulvodynia research and the efforts to minimize animal suffering would have been beneficial.

Reply:

Thank your comment. The following was added to the discussion, 4.5- selection of the animal and ethical considerations: The use of animal models in research needs to be conducted according to universal ethical concerns. All the papers discussed in this review mentioned that they followed the National Institutes of Health guidelines.

  1. Future Research Directions: Although the article concludes with promising directions for future research, it does not provide a detailed roadmap for addressing the gaps identified in the current literature. Specific recommendations for future studies, including the development of more human-like models and the exploration of novel therapeutic targets, would enhance the article's contribution to the field.

Reply:

thank you for your comment. We added the following paragraph to end of the discussion: Such studies may include: research into novel treatment with cannabinoids, various agents blocking each step of the mechanism of the development of vulvodynia, the involvement of the microbiome; Further studies on the inhibition of MCs activity and research of central nervous system involvement in the development of vulvodynia and its treatment.

  1. Lack of Interdisciplinary Perspectives: The review primarily focuses on biological and physiological aspects of vulvodynia without integrating findings from interdisciplinary research, such as psychological, social, and environmental factors that could influence the condition.

Reply:

Thank you for your comment. Indeed, the involvement of several factors including hormonal,

immunological, and genetic factors in the development of LPV has not been studied in animals yet.

This sentence has been added in the end of the discussion as a suggestion for further studies:

In addition, the involvement of other factors including hormonal, immunological, and genetic issues in the development of LPV should also be studied in animals.

  1. Inadequate Consideration of Sex and Gender Differences: The article does not address how sex and gender differences might influence the pathogenesis of vulvodynia or the response to treatments in animal models, which is crucial for developing effective and personalized interventions.

Reply:

Thank you for your comment. Since vulvodynia develops only in women, the various animal models in the literature utilized only female animals.

  1. Insufficient Analysis of Data Quality and Reproducibility: The review does not critically assess the quality and reproducibility of the data from the studies it discusses. Given the challenges of translational research from animal models to human conditions, an analysis of data robustness and reproducibility would strengthen the review's conclusions.

Reply:

Thank you for your comment. Unfortunately, the current studies available in the literature, are sparse in can not be used for analyzing the quality and reproducibility of the data. This sentence has been added to the studies in the discussion:

In addition, the data available in the literature on animal models of vulvodynia are not robust and therefore cannot be used for analyzing the quality and reproducibility of the data.

  1. PRISMA and QUADAS-2 guidelines. The review does not mention the accepted guidelines helpful to conduct a review article.

Reply:

Thank you for your comment. This review is a narrative review. The PRISMA and QUADAS-2 guidelines are not necessarily required. We have added the review description to the introduction. We edited the following sentence in the introduction:

This narrative review aims to comprehensively present and discuss the animal models employed in vulvodynia research, comparing the various induction methods and analyzing their findings.